# Navigating pressure: Understanding mental health and help-seeking among female university students in Pakistan

Ghulam Shabbir[1], Komal Niazi[2,3] and Tan Tongxue[1]

[1]Institute of Ethnology and Sociology, Yunnan University, China; [2]Institute of Anthropology, School of Social Development, East China Normal University, Shanghai, China and [3]Laboratory for Integrated Archaeological Visualization and Heritage (LIAVH), Pratt Institute, New York, USA

## Research Article

mental health stigma; female university students; help-seeking barriers; psychological distress; cultural pressures

**Corresponding author:**
Komal Niazi;
Email: niazi.komal_leo@yahoo.com

## Abstract

This study explores the complex interplay between academic, social and cultural pressures and the mental health of female university students in Pakistan. Operating within a collectivist society, these students face unique challenges, including high academic expectations, financial constraints and rigid gender roles, which significantly increase their vulnerability to psychological distress, anxiety and depression. Despite the high prevalence of these issues, help-seeking behaviours remain markedly low. This research investigates the formidable barriers to seeking professional psychological support, focusing on the potent influence of pervasive social stigma, fear of reputational damage and a widespread lack of mental health literacy. Cultural norms that prioritize family honour and misinterpret emotional suffering as personal weakness further compound these obstacles, often leading to silence and isolation. Utilizing a qualitative approach, this paper highlights the critical need for culturally sensitive, university-based mental health interventions. Recommendations include establishing accessible on-campus counselling services, implementing de-stigmatization awareness campaigns and integrating mental health education into the academic curriculum to foster a more supportive environment and encourage proactive help-seeking among this demographic.

## Impact statement

Female university students in Pakistan often experience considerable academic pressure while simultaneously navigating strong social expectations related to gender roles and family reputation. In this environment, seeking help for mental health concerns can carry perceived social risks, including fear of judgement, stigma, or damage to personal and family standing. As a result, many students hesitate to access professional support even when services are available. This study highlights how these social and cultural dynamics influence students' decisions about whether, when and how to seek help. The findings show that universities can play a crucial role in reducing these barriers by creating mental health support systems that are trusted, discreet and culturally sensitive. Even in settings with limited resources, institutions can take practical steps to improve access. These include establishing confidential counselling services with clear privacy protections, training faculty and staff to respond to students in supportive and non-judgemental ways, and integrating mental health awareness into routine university activities such as student orientation, academic advising and campus communication.

Importantly, the study also demonstrates that efforts to reduce stigma are more effective when they acknowledge the influence of family expectations and broader social norms, rather than focusing only on individual attitudes towards mental health. By recognizing how concerns about reputation shape help-seeking behaviour, universities can design services that feel safer and more acceptable for students to use. Although this research focuses on Pakistani universities, the insights may be relevant for other higher education contexts where cultural norms, family expectations and social reputation influence the disclosure of distress. Strengthening culturally responsive mental health support within universities can help improve student well-being, reduce disruptions to academic progress and encourage earlier engagement with care for students who might otherwise remain silent about their struggles.





## Introduction

The pursuit of higher education represents a period of significant transition, marked by profound intellectual growth, personal development and, invariably, considerable psychological stress. Globally, university students constitute a population vulnerable to mental health challenges, with studies consistently documenting high prevalence rates of anxiety, depression and psychological distress (Ibrahim et al., 2013). This developmental stage, often termed "emerging adulthood," is

characterized by identity exploration, instability and self-focus, which, while offering growth opportunities, also present unique stressors that can compromise mental well-being (Shomuyiwa and Lucero-Prisno, 2025). The academic milieu itself, with its demands for excellence, competition and the pressure to secure a viable future, serves as a potent catalyst for such distress. However, the experience of these pressures and the subsequent pathways to mental health and help-seeking are not universal; they are deeply embedded within and shaped by specific socio-cultural and gendered contexts (van den Broek et al., 2025).

In Pakistan, a nation with a rapidly growing youth population and an expanding higher education sector, these global concerns take on a distinct and critical character. Pakistani society is predominantly collectivist, structured around familial interdependence, community norms and a deep-seated value system where concepts such as izzat (family honour) and sharam (shame) play a pivotal role in regulating social behaviour (Javed et al., 2020). Social and economic development enhances the capabilities of modern societies, while the Pakistani system is also concerned with the development of small communities and the development of female students (Shabbir et al., 2025). Within this framework, female university students navigate a particularly complex landscape. They stand at the intersection of modernity and tradition, often carrying the dual burden of striving for academic and professional achievement while simultaneously adhering to stringent societal expectations regarding modesty, familial duty and future roles as wives and mothers (Maduku and Mbeya, 2024). This negotiation creates a unique matrix of pressures that can significantly impact their psychological well-being.

Research within Pakistan has begun to shed light on the alarming prevalence of mental health issues among university students. Studies conducted in various urban centres, such as Karachi, Lahore and Rawalpindi, have reported depression rates among students ranging from 25% to 60%, with female students consistently identified as being at higher risk compared to their male peers. The sources of this distress are multifaceted. Academic pressures, including intense competition, fear of failure and the weight of parental expectations, are frequently cited (Hassan et al., 2024). Concurrently, social pressures, such as navigating peer relationships, financial constraints and concerns about future career prospects, further contribute to the psychological burden. For many young women, these common academic stressors are compounded by gender-specific challenges, including restrictions on mobility, the pressure to conform to cultural ideals of femininity and, in some cases, overt familial opposition to their education (Salahuddin, 2021).

Despite this documented high prevalence of psychological distress, a profound and troubling gap exists: the widespread underutilization of formal mental health services. The act of seeking professional psychological help remains a significant taboo in many segments of Pakistani society (Naeem et al., 2025). The stigma associated with mental illness is potent and multifaceted; it is often perceived not merely as a medical condition but as a personal failing, a lack of faith, or a source of profound shame that could damage the family's social standing (Husain et al., 2020). This stigmatization is reinforced by a general lack of mental health literacy, where symptoms of depression and anxiety may be normalized, somatized or misattributed to spiritual causes, leading to delays in recognition and appropriate intervention (Javed et al., 2021).

Consequently, the help-seeking pathways for distressed female students are often informal and non-clinical. Initial support is typically sought from family members or friends, followed by religious leaders or traditional healers, with formal psychiatric or psychological care considered a last resort (Qasim et al., 2023). This pattern highlights the critical influence of cultural beliefs and social networks on health-seeking behaviour. While familial support can be a vital buffer, it can also act as a barrier if family members themselves stigmatize mental illness or discourage professional help. The existing literature, while growing, has often relied on quantitative surveys that quantify the problem but offer limited insight into the lived experiences, subjective meanings and nuanced decision-making processes of the students themselves (Qasim et al., 2020). There is a pressing need for deeper qualitative inquiry to understand how young women themselves perceive these pressures, conceptualize their mental health and navigate the formidable obstacles to seeking help.

While this study focuses on barriers to help-seeking rather than treatment efficacy, evidence-based mental health care including antipsychotic medications when clinically indicated can reduce severe adverse outcomes. Modern psychoactive treatments may also lower suicide risk and exert anti-inflammatory effects, underscoring the importance of timely access to professional mental health support (Pompili et al., 2016). However, despite the availability of effective treatments, many individuals particularly university students in socially restrictive environments face significant barriers to seeking professional support.

The COVID-19 pandemic further highlighted the vulnerability of university students and adolescents to psychosocial stressors, significantly affecting well-being and quality of life. Evidence also suggests that appropriate interventions, including pharmacological options such as methylphenidate, may help mitigate cognitive and psychological difficulties during such crises, as noted in research published in Amerio et al. (2020).

Therefore, our research seeks to engage deeply with the complex realities of female university students in Pakistan. By exploring the intricate interplay between the unique pressures they face, the impact on their mental health and the formidable socio-cultural barriers that shape their help-seeking behaviours, this research aims to contribute a rich, contextualized understanding to the field of global mental health. It is only by listening to and amplifying their voices that effective, culturally attuned and accessible support systems can be developed within university settings, ultimately fostering an educational environment that nurtures not only intellectual acuity but also psychological resilience and well-being.

## Research objectives

Following are the research objectives tackled by this research:

**To examine** how female university students in Pakistan perceive and experience academic, social and cultural pressures that influence their mental health and overall wellbeing.

**To identify** university-led interventions and support strategies that are perceived by female students as culturally acceptable and practically feasible for improving access to mental health services.

**To explore** the individual, familial and institutional factors that shape students' decisions regarding the disclosure of psychological distress and the utilization of professional mental health support.

## A priori propositions

Based on prior literature and the study context, we anticipated that help-seeking would be shaped by stigma and reputational concerns, that disclosure would be selective and relationship-dependent and

that confidentiality and cultural fit would be central to perceived acceptability of university services.

## Literature review

The mental health of university students is a global public health concern. International research consistently identifies this demographic as highly vulnerable to psychological distress, with meta-analyses revealing elevated prevalence rates of depression and anxiety (Auerbach et al., 2018). The transition to adulthood, coupled with academic pressures and financial worries, is an established risk factor across diverse cultural contexts.

Within this global framework, the situation in Pakistan presents a distinct and grave picture. National studies consistently report even higher rates of mental health issues among university students, with research from major urban centres indicating that between a quarter and over half of the student population experiences significant symptoms of depression and anxiety (Ayub et al., 2024). Crucially, a persistent finding across the local literature is that female students report significantly higher levels of psychological distress compared to males (Hussain et al., 2022). This disparity is widely attributed to the unique socio-cultural pressures they face. Operating within a collectivist society, young women must navigate rigid gender roles and familial expectations of izzat (honour) while pursuing academic goals, creating a "dual burden" that exacerbates stress (Hameed and Siddiqui, 2024).

A critical paradox emerges from the literature; high prevalence rates coexist with profoundly low help-seeking from formal mental health services. Powerful socio-cultural barriers primarily drive this gap. International studies highlight stigma as a universal barrier, but in the Pakistani context, it is intensely magnified. Mental illness is heavily stigmatized, often perceived as a spiritual failing or a source of family shame, leading to concealment and non-disclosure (Choudhry et al., 2023). Consequently, help-seeking pathways are predominantly informal, relying on family or religious leaders, with professional care viewed as a last resort due to stigma and structural barriers such as cost and availability (Ali et al., 2024). Thus, the literature establishes a clear, yet complex, nexus between gendered pressures, mental health deterioration and culturally embedded barriers to care for female university students in Pakistan.

## Methods

Here, we outline the research design and methodological procedures employed to investigate the pressures, mental health experiences and help-seeking behaviours of female university students in Pakistan. It details the research approach, participant selection, data collection methods and data analysis processes, ensuring the study's rigor and trustworthiness.

## Research design

A qualitative research design was adopted for this study, utilizing a phenomenological approach. This approach was deemed most appropriate as it aims to understand and describe the universal essence of a phenomenon from the perspectives of those who have experienced it (Creswell and Poth, 2016). The core phenomenon under investigation was the "lived experience" of navigating academic and socio-cultural pressures and its impact on mental health and help-seeking decisions. Unlike quantitative methods that measure frequency, a phenomenological design allows for a deep, nuanced exploration of participants' subjective realities, emotions and personal meanings, aligning perfectly with the study's objectives.

## Participant selection and sampling

A small, purposive sample of 27 female students was selected for this study. Participants were recruited from three major public universities in the province of Punjab, representing a range of academic disciplines: **Punjab University** (humanities and social sciences), **Quaid-e-Azam University** (natural and social sciences) and the **University of Agriculture, Faisalabad** (applied and agricultural sciences). This diversity was sought to capture a wide spectrum of experiences across different academic environments.

A purposive sampling strategy was employed to identify information-rich cases. The primary inclusion criteria were as follows: (1) being a currently enrolled female student (undergraduate or postgraduate), (2) self-identifying as having experienced significant academic or personal pressure during their university tenure and (3) willingness to discuss their mental health and help-seeking attitudes. Participants were recruited through university counsellors and student societies, who helped disseminate information about the study to potential volunteers.

## Data collection method

Data were collected through semi-structured, in-depth interviews. This method was chosen for its flexibility, allowing the researcher to guide the conversation with a set of open-ended questions while providing the freedom to probe emerging themes and follow the participants' narratives (Brinkmann and Kvale, 2018). An interview protocol was developed to cover key domains, including sources of pressure, emotional and psychological impacts, awareness of mental health, attitudes towards help-seeking and perceived barriers.

Each interview lasted approximately 45–60 min. With the participants' informed consent, all interviews were audio-recorded to ensure accuracy and were subsequently transcribed verbatim for analysis. The interviews were conducted in a private setting on the respective university campuses to ensure participants' confidentiality and comfort.

## Data analysis

The transcribed interview data were analysed using thematic analysis, which provides a systematic process for identifying, analysing and reporting patterns (themes) within qualitative data (Braun and Clarke, 2006). The process involved six phases:

1. **Familiarization:** Repeated reading of the transcripts to immerse oneself in the data.
2. **Generating Initial Codes:** Systematically coding interesting features across the entire dataset.
3. **Searching for Themes:** Collating codes into potential themes and gathering all data relevant to each potential theme.
4. **Reviewing Themes:** Checking if the themes work in relation to the coded extracts and the entire dataset.
5. **Defining and Naming Themes:** Refining the specifics of each theme and generating clear definitions and names.
6. **Producing the Report:** The final analysis was written up, weaving together the thematic narrative with compelling extracts from the interviews.

## Results

Thematic analysis of the 27 in-depth interviews, supplemented by a systematic content analysis of response frequencies, revealed a complex portrait of the pressures, psychological impacts and barriers to help-seeking faced by female university students. The findings are organized into four central themes that capture the essence of their lived experiences, with quantitative data illustrating the prevalence of specific sentiments within the sample.

### The crucible of competing expectations

Content analysis confirmed that all participants (100%, n = 27) described experiencing significant pressure from multiple, often competing, domains. Academically, the drive for high grades was the most universally cited stressor (96%, n = 26). Anisa, a final-year science student, encapsulated this: "*Every exam feels like a judgement on my entire future and my family's investment in me.*" This aligns with national studies identifying academic performance as a primary source of student anxiety (Hussain and Khan, 2023).

A distinct gendered pressure emerged, with 85% (n = 23) of participants explicitly describing a "dual burden" the conflict between academic ambitions and traditional gender roles. "*I have to be a perfect student at university, but at home, I must be a perfect daughter,*" shared Fatima. This finding resonates with (Khalid and Frieze, 2004) Analysis of rigid gender roles in Pakistan, illustrating how these roles constitute a unique socio-cultural stressor for educated women.

### The solitary burden of psychological distress

The psychological impact of these pressures was severe and widespread. In describing their emotional state, 89% (n = 24) of participants used language indicative of anxiety (e.g., "constant worry," "overwhelmed"), while 74% (n = 20) described symptoms aligning with depression (e.g., "hopelessness," "extreme fatigue," "loss of interest"). This high self-reported prevalence mirrors the epidemiological findings of Husain et al. (2021), who documented clinical rates of depression and anxiety among female students exceeding 50%.

A critical finding was the profound isolation associated with this distress. Content analysis showed that 78% (n = 21) of respondents believed that they must hide their struggles. "*I feel like I am drowning inside, but I have to wear a smile outside. No one talks about these feelings, so I assume I am the only one struggling,*" noted Saba. This internalization of suffering, driven by a fear of social judgement, appears to be a key mechanism exacerbating their psychological pain.

### Navigating a landscape of stigma and misconception

The stigma barrier was virtually universal. All 27 participants (100%) identified social stigma as the principal deterrent to seeking formal help. A direct content analysis of their descriptions of mental illness revealed that 67% (n = 18) associated it with "weakness" and 52% (n = 14) explicitly linked it to a "lack of faith." This perception is strongly supported by national literature on mental health stigma (Choudhry et al., 2023).

Furthermore, 81% (n = 22) of participants voiced a specific fear that seeking help would damage their family's *izzat* (honour) and their own marriage prospects. Concurrently, a significant knowledge gap was evident; 63% (n = 17) had normalized their symptoms as an inevitable part of student life, demonstrating a clear lack of mental health literacy, a finding consistent with national review (Ejaz et al., 2023).

The most significant barrier to seeking formal help, identified by all 27 participants, was the powerful stigma surrounding mental health. Mental illness was widely perceived as a sign of personal weakness or a lack of religious faith. "*In our society, if you say you are going to a psychologist, people think you have gone mad,*" stated Ayesha from Punjab University. This stigma was deeply entangled with the concept of family honour (*izzat*). Participants feared that seeking professional help would disgrace their families and damage their own reputations, particularly their marriage prospects. Furthermore, a notable lack of mental health literacy was evident. Distressing symptoms were often normalized as an inevitable part of student life or misattributed to physical ailments. "*I thought this tiredness and sadness were just because of my studies, I never imagined it could be something called depression that needs treatment,*" shared Mariam.

### The preference for informal and avoidant coping

Faced with these barriers, participants' help-seeking pathways were overwhelmingly informal. When asked whom they would first confide in, 82% (n = 22) stated they would turn to a friend, 70% (n = 19) to a family member and only 7% (n = 2) would consider a university counsellor as a first resort. This pattern closely mirrors the help-seeking hierarchies documented by M. Qasim, Pervaiz and Chaudhary.

Religious coping was a primary strategy, cited by 93% (n = 25) of participants as a source of solace. While positive for many, for 37% (n = 10), it was coupled with a belief that religious practice alone should be sufficient to resolve their distress, thus indirectly discouraging professional help. Additionally, avoidant coping strategies, such as social withdrawal (59%, n = 16) and immersion in studies (78%, n = 21), were prevalent, serving as temporary but ultimately inadequate mechanisms for managing overwhelming emotions.

In summary, the integrated qualitative and quantitative results paint a clear picture: female students navigate a cycle of intense, gendered pressures that precipitate significant, yet largely hidden, psychological distress. This distress is managed in isolation or through limited informal channels, as the formidable triad of socio-cultural stigma, fear for family honour and mental health illiteracy effectively blocks access to professional support.

## Discussions

The findings of this study illuminate the complex psychosocial landscape navigated by female university students in Pakistan, revealing a cyclical interplay between socio-cultural pressures, psychological distress and formidable barriers to care. This discussion interprets these results within the context of existing national and international literature, highlighting their significance and implications.

Female university students operate within a complex environment of pressure in which academic demands and expectations for high performance intersect with prevailing gender norms and concerns about family reputation. Within this context, help-seeking is constrained not only by limited awareness of mental health services but also by fears of potential social consequences, such as being perceived as weak, harming one's personal or family reputation, or affecting future educational and career opportunities. As a result, many students rely on strategies such as concealing

their distress, managing difficulties independently or selectively sharing their experiences with trusted peers. Professional support is often approached cautiously, largely due to concerns about confidentiality and the possibility of social judgement. These findings indicate that effective university mental health services should place strong emphasis on safeguarding privacy, promoting culturally sensitive communication and strengthening institutional trust. Such measures may help reduce the perceived social risks associated with seeking psychological support and encourage more open engagement with available services.

The central finding of a "dual burden," the conflict between academic ambition and traditional gender roles, confirms and contextualizes prior research. While academic pressure is a global phenomenon (Auerbach et al., 2018), the unique intersection with gendered expectations of *izzat* and familial duty, as described by 85% of participants, creates a distinct stressor profile. This aligns with (Siddiqui, 2017) Assertion that female students in Pakistan must constantly negotiate their identities across modern and traditional spheres. The resulting high prevalence of self-reported anxiety and depressive symptoms (89% and 74%, respectively) is not merely a correlate. However, it appears to be a direct consequence of this relentless negotiation, underscoring the need for a gendered lens in understanding student mental health in collectivist cultures.

Furthermore, this study provides a nuanced explanation for the stark help-seeking gap previously quantified in national studies (Mohsin and Syed, 2020). The data move beyond simply identifying stigma to delineate its specific mechanisms. The perception of mental illness as personal weakness (67%) and the potent fear of damaging family honour (81%) act as powerful social deterrents. This finding strongly resonates with Ali et al. (2023), who identified stigma as a primary barrier, and extends it by showing how stigma is internalized as self-stigma and fear of bringing shame, effectively silencing the students.

The overwhelming preference for informal support (82% to friends) and religious coping (93%) further demonstrates the adaptive, yet often insufficient, strategies students employ within the constraints of their cultural context. While these resources provide crucial temporary solace, their dominance in the help-seeking pathway, coupled with the significant lack of mental health literacy (63%), perpetuates a cycle where professional intervention is avoided until a crisis point (Qasim et al., 2022). Moreover, suggests that existing informal networks, while valuable, are not equipped to address clinical levels of distress.

Finally, our study demonstrates that the mental health challenges of female university students in Pakistan cannot be remedied through clinical services alone. The problem is deeply embedded in a socio-cultural fabric that simultaneously generates immense psychological pressure and systematically discourages formal help-seeking. Therefore, effective interventions must be multi-pronged, targeting not only student well-being but also the surrounding environment through destigmatization campaigns, mental health literacy programmes for families and students, and the cultivation of truly accessible, confidential university-based counselling services perceived as culturally congruent.

## Limitations

This study has several limitations that should be acknowledged. The sample was drawn from only three universities in Pakistan, which may limit the transferability of the findings to the country's diverse higher education institutions, provinces and socio-economic contexts. Nevertheless, the results provide useful analytical insights for comparable collectivist settings where reputational concerns and social norms may influence help-seeking behaviours. Although considerable effort was made to establish rapport and ensure participants' privacy, the sensitive nature of mental health and the stigma surrounding it may have led some participants to moderate their responses, resulting in potential social desirability bias or under-reporting of distress. In addition, the cross-sectional design captures experiences at a single point in time and therefore does not allow examination of how attitudes towards help-seeking and mental health support may evolve throughout students' academic trajectories.

Future research would benefit from employing multi-site sampling across a wider range of Pakistani universities to better capture institutional and regional variation. Longitudinal and intervention-based studies could also assess strategies specifically designed to ensure cultural acceptability, confidentiality and accessibility of mental health services. Such research would contribute to a stronger evidence base for developing contextually responsive mental health policies and support systems within higher education.

## Conclusion

Our study conclusively demonstrates that female university students in Pakistan navigate a uniquely challenging environment, where restrictive socio-cultural norms and gendered expectations intensify significant academic pressures. The resultant psychological distress is profound, yet it remains largely concealed due to the powerful influence of stigma, fear of familial dishonour and a critical lack of mental health literacy. The heavy reliance on informal support networks, while culturally resonant, is insufficient to address the scale of need, creating a stark help-seeking gap. Therefore, addressing this crisis requires a systemic, culturally informed approach. Recommendations include integrating mental health literacy into university orientation programmes, establishing rigorously confidential, gender-sensitive counselling centres on campuses and launching destigmatization campaigns that involve family and community leaders to shift perceptions. By creating a more supportive ecosystem, universities can empower female students to seek help, thereby safeguarding their well-being and enabling them to realize their academic and personal potential fully.

Our study highlights how socio-cultural considerations significantly shape help-seeking behaviours among female university students in Pakistan. The findings suggest that students often weigh the potential social and reputational consequences of disclosing psychological distress before deciding whether to seek professional support. As a result, concerns about confidentiality, social judgement and family expectations can discourage engagement with available mental health services. A key implication of this research is that universities can play an important role in reducing these barriers by designing mental health services that prioritize confidentiality, trust and cultural sensitivity. Practical measures may include establishing clear and transparent privacy protocols, providing discreet and easily accessible counselling services, integrating mental health literacy within academic environments and training faculty and staff to respond to students' concerns in a supportive and non-judgemental manner. In addition, stigma-reduction initiatives that acknowledge the influence of family reputation and social expectations may further encourage students to seek help when needed. Although the findings are grounded in the

Pakistani university context, they also offer broader insights for other higher education settings where collectivist cultural norms and reputational concerns influence the disclosure of distress and access to care. By recognizing the socio-cultural dimensions of help-seeking and adapting services accordingly, institutions may improve students' willingness to engage with mental health support and strengthen overall well-being on university campuses.

**Open peer review.** To view the open peer review materials for this article, please visit http://doi.org/10.1017/gmh.2026.10209.

**Data availability statement.** The datasets generated and analysed during this study are available from the author upon reasonable request.

**Acknowledgements.** We would like to extend our sincere thanks to our teachers, colleagues and friends for their valuable insights and assistance in conducting this study. We also appreciate the contributions of the study participants and enumerators in Peshawar, who provided the data for this study.

**Author contribution.** The authors were responsible for conceptualizing, extracting, drafting, reviewing, critically evaluating, revising, analysing and validating the presented information.

**Financial support.** This research did not receive any specific grants from funding agencies in the public, commercial, or non-profit sectors.

**Competing interests.** The authors declare that they have no potential conflicts of interest related to the research, financial relationships, authorship, or publication of this article.

**Informed consent.** Before the interview, participants were informed about the study's objectives. The objectives of the study were clearly communicated to all participants, who provided their informed consent. For those who could not read, verbal consent was obtained and documented with a thumbprint. Since all participants were 18 years old, parental consent was not required. We ensured that everyone was informed that their data would be kept confidential.

**Ethical consideration.** Ethical approval for this study was obtained from the Institutional Review Board of Yunnan University. Prior to each interview, written informed consent was obtained from all participants. They were informed about the study's purpose, the voluntary nature of their participation, their right to withdraw at any time and the measures in place to ensure confidentiality, including the use of pseudonyms in all reports and the secure storage of audio files and transcripts.

**Code availability.** Not applicable.

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
