## [Reviewer Report]

The manuscript titled “Navigating Pressure: Understanding Mental Health and Help-Seeking among Female University Students in Pakistan” offers an insightful and timely contribution to the field of global mental health. The authors effectively highlight the multifaceted pressures experienced by female university students within a collectivist Pakistani context, emphasizing the intersection of academic demands, financial limitations, gendered expectations, and cultural norms. The qualitative approach is well-justified and allows for a nuanced exploration of students’ lived experiences, particularly regarding stigma, reputational concerns, and limited mental health literacy—all of which significantly hinder help-seeking behavior. The manuscript is grounded in relevant literature and demonstrates strong analytical depth, clearly articulating the role of cultural sensitivity in understanding mental health vulnerabilities. Importantly, the paper moves beyond problem identification to offer practical and contextually appropriate recommendations, including the development of university-based counselling services, destigmatization campaigns, and mental health education integrated within academic structures. These suggestions are both feasible and aligned with ongoing global efforts to enhance equitable access to psychological support. Overall, the manuscript is well-written, methodologically sound, and highly relevant to the journal’s scope. I recommend its acceptance, as it contributes meaningfully to discussions on prevention, intervention, and culturally informed care in low-resource settings.

---

## [Reviewer Report]

Thank you very much for asking me to review the present paper.

This is, in summary, an interesting study aimed to explore the complex interplay between academic, social, and cultural pressures and the mental health of female university students in Pakistan. The authors stressed the formidable barriers to seeking professional psychological support, focusing on the potent influence of pervasive social stigma, fear of reputational damage, and a widespread lack of mental health literacy. Importantly, cultural norms that prioritize family honour and misinterpret emotional suffering as personal weakness further compound these obstacles, often leading to silence and isolation. Finally, utilizing a qualitative approach, this paper highlights the critical need for culturally sensitive, university-based mental health intervention.

The authors may find as follows my main comments/suggestions as follows.

First, when throughout the Introduction section, as throughout the same section, the authors adequately stressed the importance of psychological stress, they might further stress the role of antipsychotics in attenuating negative clinical outcomes and psychological stress. Importantly, there are evidence supporting for potential suicide risk-reducing effects and anti-inflammatory activities of modern psychoactive treatments. Thus, according to this background, the article published on Int J Mol Sci in 2016 (PMID: 27727180) might be briefly cited within the main text.

In addition, when throughout the same section, the authors stressed that the importance and impact of alarming prevalence of mental health issues among university students even in difficult periods, they might even briefly refer to the important consequences related to Covid-19 pandemic and the potential of methylphenidate in order to attenuate difficult situations in adolescents. Importantly, they further stress that individual wellbeing and quality of life have been significantly related to increased psychosocial issues during Covid-19, which enhanced the vulnerability of patients, in particular those with cognitive dysfunctions, to stressful situations. Thus, according to this background, the article published on Acta Biomed (PMID: 32701924) might be briefly cited within the main text.

Importantly, as the main aims/objectives of this study have been correctly reported, the most relevant hypotheses might be similarly reported.

Furthermore, the most relevant limitations/shortcomings of the present study could be reported more extensively and comprehensively within the main text.

Notably, the authors could immediately present and discuss, in the first lines of the Discussion section, the most relevant study findings of this paper instead of focusing on the most relevant literature findings regarding the main topic that should have been adequately stressed elsewhere within the main text.

Finally, what is the take-home message of this manuscript? While the authors stressed that female university students in Pakistan navigate a uniquely challenging environment where restrictive socio-cultural norms and gendered expectations intensify significant academic pressures, the conclusive remarks of this study might be provided in a more detailed manner for the readers. What are, specifically, the main implications of these findings and how the present results may be generalized? Here, some additional information could be useful for the general readers.

---

## [Editor Report]

Thank you for submitting your manuscript, “Navigating Pressure: Understanding Mental Health and Help-Seeking among Female University Students in Pakistan”, to Cambridge Prisms: Global Mental Health. Please revise your manuscript according to the reviewers’ comments, and resubmit it for further consideration.

---

## [Reviewer Report]

Thank you for asking me to review again the present paper. In the revised manuscript, the authors addressed most of the major questions raised by Reviewers improving both the main structure and quality of the present paper. I have no further additional comments.

---

## [Editor Report]

Thank you for submitting your revision of the manuscript titled: “Navigating Pressure: Understanding Mental Health and Help-Seeking among Female University Students in Pakistan.” The revisions successfully address all points raised during the review process, and the manuscript is now suitable for publication in its current form.